# Study on the Coupling Relationship of Low Temperature Fluidity and Oxidation Stability of Biodiesel

**Shuaishuai Lv [1], Jiaqiao Zhang [1,2], Hongjun Ni [1,*], Xingxing Wang [1], Yu Zhu [1] and Lei Chen [1]**

[1] College of Mechanical Engineering, Nantong University, Nantong 226019, China; lvshuaishuai@ntu.edu.cn (S.L.); c501937020@tokushima-u.ac.jp (J.Z.); wangxingxing.ntu@gmail.com (X.W.); zhuyu.ntu@gmail.com (Y.Z.); chenlei.ntu@gmail.com (L.C.)
[2] Graduate School of Advanced Technology and Science, University of Tokushima, Tokushima 770-8506, Japan
\* Correspondence: ni.hj@ntu.edu.cn

**Abstract:** Low temperature fluidity and oxidation stability are important indicators for the measurement of the performance of biodiesel, which are currently two major issues in association with the use of biodiesel on diesel engines. In the current work, actors affecting the low temperature fluidity and oxidation stability of biodiesel, such as adding reagents, changing the blending ratio, were studied separately. Then, the influencing factors were comprehensively analyzed to simultaneously improve the low temperature fluidity and oxidation stability of biodiesel through adjusting the proportion of fatty acid methyl esters (FAMEs). The results show that the cold fluidity improver (CFI) exerts the greatest influence on the biodiesel blending oil B10. When the CFI is added to 0.6%, the cold filter plugging point (CFPP) of B10 is reduced to a minimum of −17 °C. Additionally, blending ratio also has a great influence on the CFPP of biodiesel blended fuel. When the amount of biodiesel added is 5%, the CFPP of biodiesel blended fuel is equivalent to the CFPP of 0 petrol diesel (0PD). When the amount of biodiesel added exceeds 50%, the oxidation induction time (OIT) of biodiesel with different blending ratios can be made greater than 6 h by adding butylated hydroxyanisole (BHA) with a ratio of 0.1%. The CFPP and OIT of the blended fuel increase with the increasing of PME addition ratio. When the blending ratio of palm oil methyl ester (PME) and rapeseed oil methyl ester (RME) is R60P40, the CFPP is 0 °C, and the OIT is 5.9 h.

**Keywords:** biodiesel; low temperature fluidity; oxidation stability; coupling relationship

## 1. Introduction

With the rapid development of the economy, the economic development based on petrochemical energy is increasingly constrained by the petrochemical resources shortage and environmental pollution. As a substitute for petrochemical diesel, biodiesel has been extensively concerned by countries globally for its environmental friendliness and renewable advantages with a very broad market prospect [1,2].

The higher the content of saturated fatty acid methyl ester in biodiesel, the easier it is to crystallize and precipitate at a low temperature, affecting the storage and transportation of biodiesel at a low temperature, and making it easier to block the pipes and filters of diesel engines, while in turn influencing their normal oil supply [3–7]. The higher the content of unsaturated fatty acid methyl ester in biodiesel, the worse the oxidation stability is. Meanwhile, it is more likely to oxidize during the storage period, and it is not suitable for long-term storage. Moreover, the biodiesel contains unstable double bonds and will be polymerized in the oil pipeline for long-term use. The formation of macromolecular gelatinous substances, such as methyl dimer acid, may cause problems such as fuel system gelation, filter and nozzle clogging [8–10]. Obviously, the low temperature fluidity and

oxidation stability of biodiesel not only affect its performance, but also affect the operation of various systems of motor vehicles and the storage of biodiesel [11–13]. As a result, the low temperature fluidity and oxidation stability of biodiesel are investigated. The performance of biodiesel is objectively evaluated, and it provides certain guiding significance for screening biodiesel additives. Additionally, it can promote the application of biodiesel on diesel engines and alleviate energy crises, thus improving the ecological environment.

At present, the research on low temperature fluidity mainly focuses on the influencing factors of low temperature fluidity of biodiesel, the development of cold fluidity improver and the methods of improving low temperature fluidity. The research on oxidation stability has mainly focused on the factors affecting the oxidation stability of biodiesel, the oxidation mechanism and the development and screening of biodiesel antioxidants. However, the low temperature fluidity and oxidation stability of biodiesel are associated with each other. Few people have comprehensively studied the low temperature fluidity and oxidation stability of biodiesel; that is, the coupling relationship between the two. However, there is a mutual coupling relationship between low temperature fluidity and oxidation stability of biodiesel. How to simultaneously improve the low temperature fluidity and oxidation stability of biodiesel remains a difficult problem in biodiesel research.

Bryan R.M. [14] studied the low temperature fluidity of biodiesel blending oil, finding that the biodiesel fatty acid composition will exert no effect on the low temperature fluidity of low-proportion biodiesel blending oil unless biodiesel the content of medium and long carbon chain saturated fatty methyl ester is >48%. Wu M.X. [15] and other scholars believed that biodiesel fatty acid methyl ester composition is one of the factors which can affect its low temperature fluidity. The low temperature fluidity of biodiesel deteriorates with the increasing of saturated fatty acid methyl ester content and carbon chain length. Ramos M.J. et al. [16] considered that the oxidation stability of biodiesel is related to raw materials. Vegetable oil is abundant in polyunsaturated linoleic acid and linolenic acid, making the biodiesel produced by it generally poor in oxidation stability. James P. et al. [17] concluded that the oxidation stability of biodiesel relates to the unsaturation of biodiesel fatty acid methyl ester, and the antioxidant used are caffeic acid and ferulic acid. The results demonstrate that both antioxidants can effectively delay the storage time of the oxidation process at the initial time, and the antioxidant effect of caffeic acid is stronger than that of ferulic acid. Lv Y. et al. [18] enhanced the low temperature flow performance through the blending of biodiesel from different sources. It is considered that for some high-pour point biodiesel, the effect of improving the low temperature flow performance through oil blending is more effective in comparison with adding petrochemical diesel pour point depressant.

In the current work, cold fluidity improver and antioxidants were added to biodiesel to investigate the effects of the blending ratio on biodiesel performance. Furthermore, ten kinds of biodiesel fatty acid methyl esters were selected to study the influence of their composition on the coupling relationship between low temperature fluidity and oxidation stability of biodiesel, as well as to explore how to simultaneously improve the low temperature fluidity and oxidation stability of biodiesel.

## 2. Materials and Methods

### 2.1. Experiment Materials

#### 2.1.1. Low Temperature Fluidity Experiment Materials

The test oil sample is palm oil methyl biodiesel (PME), laboratory-made, which is in consistence with GB/T 20828-2007; 0 petrol diesel (0PD), produced by Sinopec, Beijing, China.

The cold fluidity improver (CFI) is Bangjie diesel anti-coagulant, produced by Sinopec, Beijing, China. Its main compositions include derivatives of olefin polymers and ethylene-vinyl acetate copolymers.

### 2.1.2. Oxidation Stability Experiment Materials

The test oil sample is rapeseed oil methyl biodiesel (RME), laboratory-made, in line with GB/T 20828-2007; 0 petrol diesel (0PD), produced by Sinopec, Beijing, China.

Table 1 presents the reagents used in the experiment.

**Table 1.** Main reagents used in the experiment.

| Reagent Name | Manufacturer |
| --- | --- |
| BHT (2,6-di-tert-butyl-4-methylphenol) | Shanghai Pengxiang Chemical Co., Ltd. (Shanghai, China) |
| BHA (2-tert-butyl-4-methoxyphenol) | Shanghai Pengxiang Chemical Co., Ltd. (Shanghai, China) |
| TBHQ (Tert-butyl hydroquinone) | Zhongshan Jiahui Food Additive Co., Ltd. (Zhongshan, China) |
| PG (3,4,5-trihydroxybenzoic acid) | Shanghai Yanxintang Biological Technology Co., Ltd. (Shanghai, China) |

### 2.2. Experimental Methods

#### 2.2.1. Low Temperature Fluidity Experiment Methods

Produced by Jinshi City Petrochemical Instrument Co., Ltd., Hunan, China, the experimental device was civil fuel cold filter plugging point tester.

Initially, 45mL of biodiesel sample was poured into the test cup, and the sample was heated to $30 \pm 5\,°C$ by water bath. Then, the sample was cooled. When the sample was cooled to 5–6 °C higher than the expected CFPP, suction was performed at a pressure of 1.96 kPa. In addition, the suction stopped when the sample passed through a filter 20 mL. Subsequently, the sample was cooled at 1 °C intervals and suck. Repeat this operation until the sample passing through the filter was less than 20 mL in 60 s. Record the temperature at this time, which is the Could Filter Plugging Point (CFPP) of the sample.

#### 2.2.2. Oxidation Stability Experiment Methods

Switzerland Metrohm 873 biodiesel oxidation stability tester is employed as the test instrument. Weigh 7.5 g ± 0.01 g of the prepared biodiesel into a sample tube and check if the depth of the air tube is insufficient in the sample. Otherwise, the sample must be replenished. Check if the sample tube heating hole is clean, blow out the dust in the hole with nitrogen and clean the measuring cell and accessories with distilled water. Adjust the heating module to the set temperature (110 °C) to keep the sample temperature at ±0.1 °C. After reaching the set temperature, the temperature display "TEMPERATURE" stops flashing and lights up, and the button turns green. At this time, the heating block gets ready, and the sample can be placed to start measurement. The clean gas is introduced into the sample reaction vessel at a flow rate of 10 L/h, and the method parameters can be modified through opening the Live Parameters window during the measurement. Add 60 mL of distilled water to the measuring cell, check the electrode, adjust the electrode signal, take a conductivity point every 20 s, and set the conductivity to a maximum of 200 us/cm.

## 3. Experiment Results and Discussion

### 3.1. Low Temperature Fluidity Experiment Results

#### 3.1.1. Effects of CFI on Low Temperature Fluidity of Biodiesel

Figures 1–4 were obtained by adding cold fluidity improver (CFI) to the biodiesel blended fuels of B10, B20, B50 and B100 [19]. In Bx, the number x represents the added percentage of biodiesel.

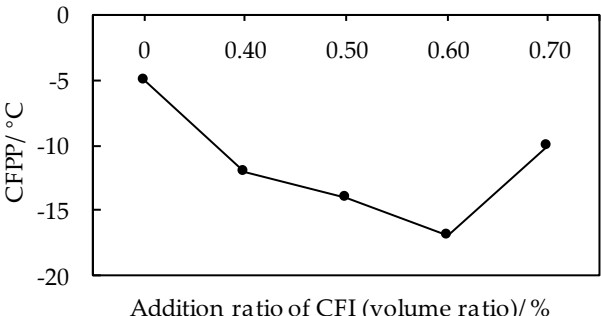

**Figure 1.** Effects of cold fluidity improver (CFI) on B10.

According to Figure 1, we can see that the biodiesel blending fuel B10 is greatly affected by the addition ratio of CFI. The CFPP of B10 is −5 °C without any CFI However, with the increasing addition of CFI, the CFPP of B10 decrease. When the added amount of CFI reaches 0.6%, the CFPP of B10 is minimized with −17 °C. Then, with the addition of increasing CFI is, the CFPP of B10 rises.

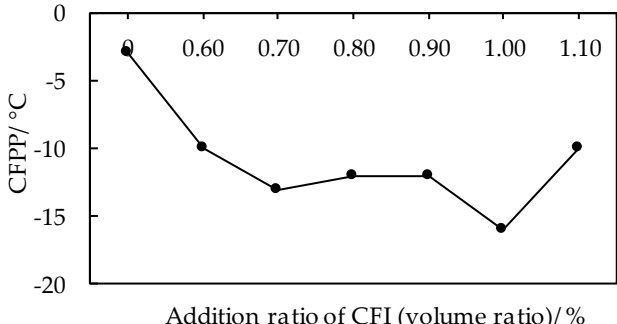

**Figure 2.** Effects of CFI on B20.

Figure 2 shows that the influence of CFI on B20 is smaller than B10. The CFPP of B20 is −3 °C when the addition ratio of CFI is 0. As the addition of CFI increases, the CFPP of B20 shows a downward trend. When CFI was added at 1%, the CFPP dropped to a minimum of 16 °C. Afterwards, as the ratio of CFI increases, CFPP begin to rise rapidly.

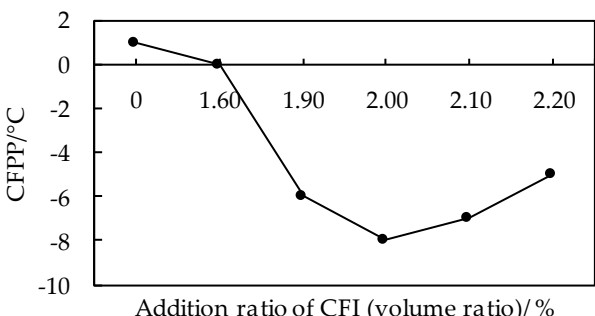

**Figure 3.** Effects of CFI on B50.

Figure 3 shows that compared with B10 and B20, the effects of CFI on the B50 is not great, and the proportion of the addition amount of CFI is significantly increased. When the addition ratio of CFI is 0, the CFPP of B50 is 1 °C. The value of CFFP presents a downward trend and then an upward trend overall. When the addition ratio is 2%, the CFFP reaches a minimum of −8 °C.

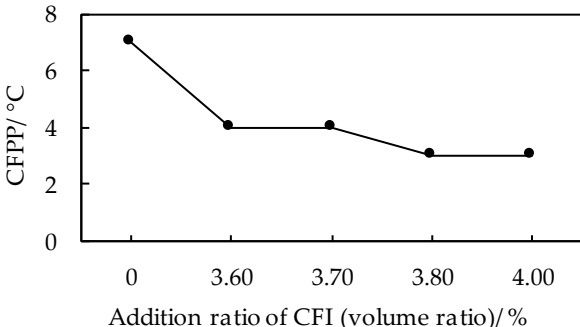

**Figure 4.** Effects of CFI on B100.

Figure 4 shows that the CFI exerts less influence on B100, and the proportion of CFI added amount is remarkably increased. The CFPP of B100 is 7 °C without any CFI. Different from other biodiesel blended fuel, as the addition ratio of CFI increases, the CFPP of B100 shows a downward trend, without increasing. When the CFI is added at 3.6%, the CFPP of B100 decreases the most, down to 4 °C, and then added the CFI, the CFPP of B100 tends to remain unchanged.

### 3.1.2. Effects of Blending Ratio on Low Temperature Fluidity of Biodiesel

Figure 5 shows that the CFPP of the biodiesel blended oil is affected by blending ratio greatly. The CFPP of B0 is −9 °C when the blending ratio is 0, and the value of CFPP keeps rising with the increase of the blending ratio. When the blending ratio is 5%, the CFPP of the blended fuel is equivalent to the CFPP of the 0PD. Furthermore, as the content of biodiesel increases, when the blending ratio is 50%, the cold filter point of the fuel has exceeded 0 °C.

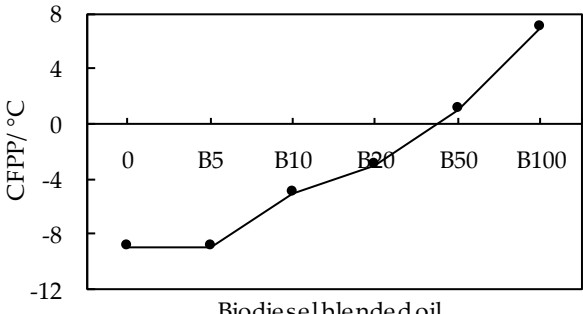

**Figure 5.** Effects of blending ratio of biodiesel blended oil on CFPP (cold filter plugging point).

### *3.2. Oxidation Stability Experiment Results*

#### 3.2.1. Effects of Antioxidants on OIT of Biodiesel

The OIT of biodiesel is influenced greatly by antioxidant, and the antioxidant effects of different antioxidants are also different. Commonly used antioxidants include BHA, BHT, TBHQ and PG. In the case of biodiesel without oxidation, the OIT was 3.72 h. After the addition of four antioxidants, BHA, BHT, TBHQ and PG, the OIT significantly increased, and both reached the national standard requirement of more than 6 h. Among the four antioxidants, BHA has the worst antioxidant effects, and TBHQ has the best antioxidant effects.

#### 3.2.2. Effects of Blending Ratio on OIT of Biodiesel

According to Figure 6, the OIT of biodiesel can be affected by the blending ratio. In the case that the biodiesel addition ratio is 10%, the OIT is 38.23 h. When the biodiesel addition ratio is 20%, the OIT

is significantly decreased. When the biodiesel addition ratio exceeds 50%, the OIT cannot satisfy the national standard requirement of more than 6 h.

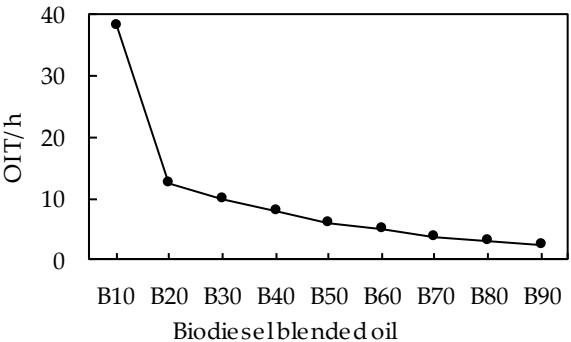

**Figure 6.** Effects of blending ratio of biodiesel on OIT (oxidation induction time).

From Figure 7, when the biodiesel addition ratio exceeds 50%, the OIT cannot meet the national standard requirement of more than 6 h. However, when the antioxidant BHA is added in a ratio of 0.1%, the OIT of B50, B60, B70, B80 and B90 in Figure 7 can reach the national standard requirement of more than 6 h. Obviously, the antioxidant exerts a significant effect on the antioxidation of biodiesel.

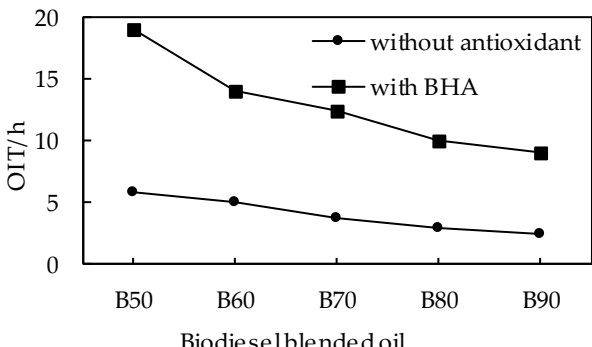

**Figure 7.** Combined effects of blending ratio and antioxidants on OIT.

*3.3. Experiment Results of Coupling Relationship between Low Temperature Fluidity and Oxidation Stability*

3.3.1. Coupling Relationship between Low Temperature Fluidity and Oxidation Stability of Biodiesel

According to the main influencing factors of low temperature fluidity and oxidation stability of biodiesel, these two properties are associated with the composition and content of FAME of biodiesel. The higher the content of long chain saturated FAME in biodiesel, the worse the low temperature fluidity [20,21]. The higher the content of unsaturated FAME, the worse the oxidation stability. As shown in Figures 8 and 9, with the increasing content of saturated FAME and unsaturated FAME, the low temperature fluidity and oxidation stability of biodiesel are mutually restricted. When the low temperature fluidity is good, the oxidation stability is poor, and vice versa. Consequently, the low temperature fluidity and oxidation stability of biodiesel have a mutually restrictive coupling relationship. Through adjusting the composition and content of FAME, the low temperature fluidity and oxidation stability of biodiesel can be simultaneously improved.

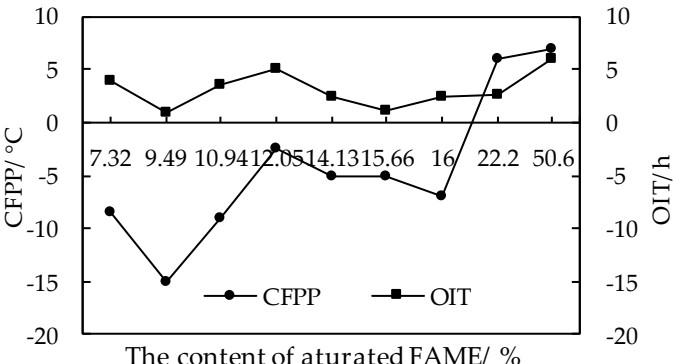

**Figure 8.** Effects of saturated FAME (fatty acid methyl ester) on low temperature fluidity and oxidation stability of biodiesel.

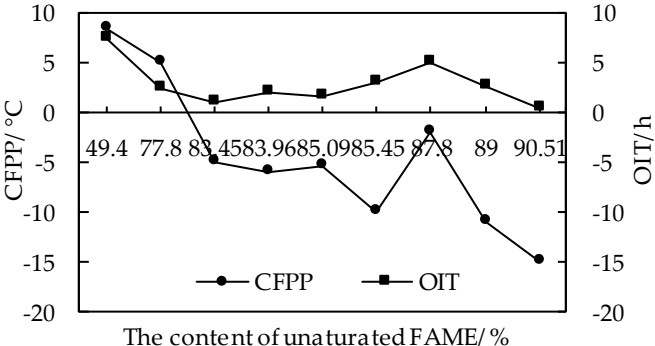

**Figure 9.** Effects of unsaturated FAME on low temperature fluidity and oxidation stability of biodiesel.

### 3.3.2. Biodiesel Fatty Acid Methyl Ester Data Acquisition

Oxidation stability and CFPP in biodiesel fuel characteristics are mainly determined by the quality of the feedstock oil, and vary little with the processing process. Consequently, the data for the composition of the 10 biodiesel fatty acid methyl esters in the current research are all from the references [12,22–24]. The ten biodiesels are rapeseed oil methyl ester (RME), tea oil methyl ester (TME), soybean oil methyl ester (SME), olive oil methyl ester (OME), peanut oil methyl ester (PNME), jatropha curcas oil biodiesel (JCME), cottonseed oil methyl ester (CSME), Chinese tallow tree seed oil methyl ester (CTKME), corn oil methyl ester (CME) and palm oil methyl ester (PME). Table 2 presents the composition of biodiesel fatty acid methyl ester. For the Cm in the table, n-m represents the number of carbon atoms of the fatty acid group and n stands for the number of double bonds of the fatty acid group. SFAME is the abbreviation for saturated fatty acid methyl ester, SUFAME is the abbreviation for unsaturated fatty acid methyl ester, and PUFAME is the abbreviation for polyunsaturated fatty acid methyl ester. Table 3 shows the ten biodiesel's CFPP and OIT.

**Table 2.** Composition of biodiesel fatty acid methyl ester.

| FAME | RME | TME | SME | OME | PNME | JCME | CSME | CTKME | CME | PME |
|---|---|---|---|---|---|---|---|---|---|---|
| Methyl laurate (C 12:0) | | | | | | | | | | 0.26 |
| Methyl myristate (C 14:0) | | | | | | | 0.40 | | | 1.09 |
| Methyl palmitate (C 16:0) | 5.54 | 8.54 | 11.35 | 11.20 | 10.90 | 5.77 | 20.40 | 7.21 | 12.14 | 44.81 |
| Methyl stearate (C 18:0) | 1.78 | 1.78 | 4.31 | 1.96 | 2.70 | 5.73 | 1.40 | 2.28 | 1.99 | 4.09 |
| Methyl aramate (C 20:0) | | 0.62 | | 2.45 | 1.10 | 0.55 | | | | 0.35 |
| Methyl behenate (C 22:0) | | | | 0.39 | 1.70 | | | | | |
| Methyl oleate (C 24:0) | | | | | 0.60 | | | | | |
| Methyl myrcene (C 12:1) | | | | | | | | 3.16 | | |
| Methyl oil palmitate (C 16:1) | | | | | | | 0.30 | | | 0.20 |
| Methyl oleate (C 18:1) | 39.96 | 80.10 | 23.25 | 71.44 | 46.50 | 20.49 | 15.10 | 14.61 | 29.49 | 39.99 |
| Methyl eicosenoate (C 20:1) | | | | 0.54 | 0.70 | 0.98 | | | | |
| Methyl erucate (C 22:1) | 16.18 | | | | 0.30 | | | | | |
| Methyl linoleate (C 18:2) | 22.35 | 8.50 | 53.58 | 10.60 | 35.40 | 66.11 | 62.40 | 31.72 | 55.03 | 8.94 |
| Methyl linolenate (C 18:3) | 6.96 | 0.40 | 6.62 | 1.38 | 0.10 | 0.22 | | 41.02 | 0.57 | 0.27 |
| Total | 92.77 | 99.94 | 99.11 | 99.96 | 100.00 | 99.85 | 100.00 | 100.00 | 99.92 | 100.00 |
| SFAME | 7.32 | 10.94 | 15.66 | 16.00 | 17.00 | 12.05 | 22.20 | 9.49 | 14.13 | 50.60 |
| SU FAME | 56.14 | 80.10 | 23.25 | 71.98 | 47.50 | 21.47 | 15.40 | 17.77 | 29.49 | 40.19 |
| PU FAME | 29.31 | 8.90 | 60.20 | 11.98 | 35.50 | 66.33 | 62.40 | 72.74 | 55.60 | 9.21 |

**Table 3.** Cold filter plugging point and oxidation induction time of biodiesel.

| | RME | TME | SME | OME | PNME | JCME | CSME | CTKME | CME | PME |
|---|---|---|---|---|---|---|---|---|---|---|
| CFPP/°C | −9 | −9 | −5 | 8 | 12 | −3 | 6 | −14 | −5 | 8 |
| OIT/h | 3.72 | 3.57 | 2.46 | 3.68 | 2.79 | 5.07 | 3.22 | 1.47 | 2.48 | 7.38 |

3.3.3. Effects of Fatty Acid Methyl Ester Composition on the Coupling Relationship between Low Temperature Fluidity and Oxidation Stability of Biodiesel

The low temperature fluidity and oxidation stability of biodiesel are closely related to the fatty acid methyl ester composition. Through plotting the three influence factors SFAME, SUFAME and PUFAME as triangle vertices, the relationship between low temperature fluidity and oxidation stability of biodiesel and fatty acid methyl esters can be obtained. As shown in Figure 10, the CFPP and OIT of biodiesel can be predicted through combining this figure and the composition of biodiesel fatty acid methyl ester.

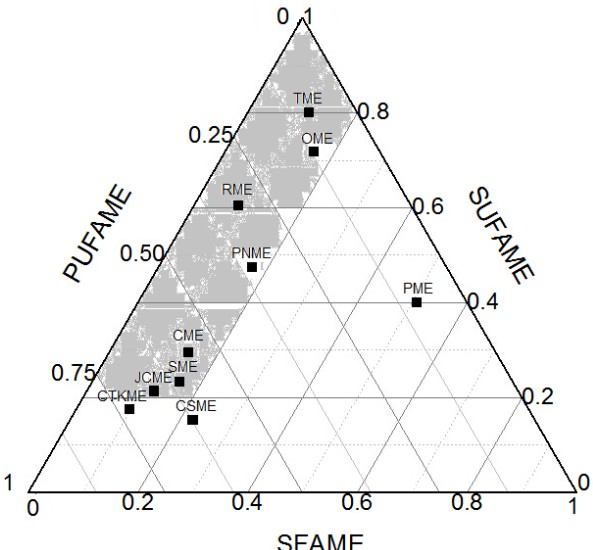

**Figure 10.** Relationship between low temperature fluidity and oxidation stability of biodiesel and fatty acid methyl esters.

It can be seen from Figure 10 that the species that meet or approach the biodiesel oxidation stability are far away from the apex of polyunsaturated fatty acids (lower left apex). Besides, the species that meet or approach the biodiesel CFPP are far from the apex of saturated fatty acids (lower right apex). Therefore, considering the biodiesel CFPP and oxidation stability requirements, the area of the gray part of the figure is delineated. The fatty acid methyl ester composition in this region generally contains a small amount of saturated fatty acids (<20%), a large amount of unsaturated fatty acids (>20%), and a certain amount of polyunsaturated fatty acids (<70%). Biodiesel in this area is JCME, SME, CME, PNME, RME, OME and TME. The main feature of these biodiesels is the high monounsaturated fatty acid methyl ester content. The only special case is PNME. The reason is that the high content of long chain saturated fatty acid methyl esters in PNME results in a high CFPP. Therefore, in the present study, not only the fatty acid methyl ester composition but also the structure and chain length of the fatty acid methyl ester should be considered.

### 3.3.4. Effects of Blending Ratio on the Coupling Relationship between Low Temperature Fluidity and Oxidation Stability of Biodiesel

From Figure 11, it can be concluded that the fatty acid methyl ester composition generally contains a small amount of saturated fatty acids (<20%), a large amount of unsaturated fatty acids (>20%), and a certain amount of polyunsaturated fatty acids (<70%). To confirm this conclusion, PME with better oxidation stability was blended with RME with better low temperature fluidity. The composition of fatty acid methyl esters of blended fuel is shown in Table 4, and the effects of blending ratio on the coupling relationship between low temperature fluidity and oxidation stability of biodiesel can be found in Figure 11.

**Table 4.** Composition of fatty acid methyl esters of blended fuel.

| Ratio | Categories | | |
|---|---|---|---|
| | SFAME | SUFAME | PUFAME |
| RME | 7.32 | 56.14 | 29.13 |
| R90P10 | 11.648 | 54.545 | 27.138 |
| R80P20 | 15.976 | 52.95 | 25.146 |
| R70P30 | 20.304 | 51.355 | 23.154 |
| R60P40 | 24.632 | 49.76 | 21.162 |
| R50P50 | 28.96 | 48.165 | 19.17 |
| R40P60 | 33.288 | 46.57 | 17.178 |
| R30P70 | 37.616 | 44.975 | 15.186 |
| R20P80 | 41.944 | 43.38 | 13.194 |
| R10P90 | 46.272 | 41.785 | 11.202 |
| PME | 50.6 | 40.19 | 9.21 |

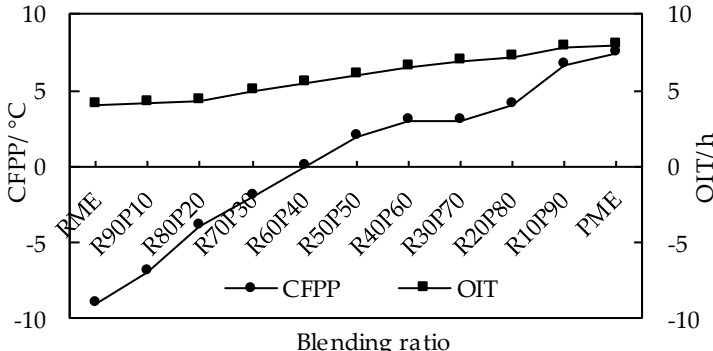

**Figure 11.** Effects of blending ratio on coupling relationship between low temperature fluidity and oxidation stability of biodiesel.

It can be seen from Figure 11 that the CFPP of the blended fuel increases with the increase of the PME addition ratio, and the low temperature fluidity gradually deteriorates. The OIT of the blended fuel increases with the increasing of the PME addition ratio, and the oxidation stability gradually becomes better. When the blending ratio of PME and RME is R60P40, the CFPP is 0 °C, and the OIT is 5.9 h. Obviously, the method for improving the low temperature fluidity and oxidation stability of biodiesel by adjusting the composition and content of biodiesel fatty acid is feasible.

## 4. Conclusions

When the biodiesel addition ratio increases, the value of CFPP of biodiesel blending fuel increases accordingly. Although the biodiesel blending oil (B10, B20, B50) has the best point for the addition of CFI, but the CFPP of biodiesel (B100) is not affected by CFI.

Biodiesel (B100) has poor oxidation stability, but it can satisfy the national standard for not less than 6 h by adding a small amount (0.1%) of antioxidants. The combination of low proportion (B10) and 0PD can effectively improve the oxidation stability. However, the oxidation stability of biodiesel blended oil is worse with the increase of the ratio of blending with 0PD. Biodiesel blending oil with a blending ratio of more than 50% can no longer meet national standards. Nevertheless, national standards can be achieved by adding a small amount (0.1%) of antioxidants.

The fatty acid methyl ester composition generally contains a small amount of saturated fatty acids (<20%), a large amount of unsaturated fatty acids (>20%), and a certain amount of polyunsaturated fatty acids (<70%). The low temperature fluidity and oxidation stability of biodiesel have a mutually constraining coupling relationship. It is feasible to simultaneously improve the low temperature

fluidity and oxidation stability of biodiesel through adjusting the composition and content of biodiesel fatty acid.

## 5. Patents

One China invention patents (Biodiesel preparation device and preparation method, CN103060098B) related to this paper have been granted.

**Author Contributions:** Conceptualization, H.N. and Y.Z.; resources, H.N. and S.L.; validation, J.Z. and L.C.; formal analysis, H.N. and S.L.; investigation, J.Z. and X.W.; writing—original draft, J.Z.; methodology, S.L.; writing—review and editing, H.N. and S.L.; supervision, H.N.; visualization, J.Z. All authors have read and agreed to the published version of the manuscript.

**Funding:** This research was supported by A Priority Academic Program Development of Jiangsu Higher Education Institutions (PAPD), Ministry of Education Industry-University Cooperation Collaborative Education Project (201901189019); Jiangsu Province University Students Innovation Training Program Key Project (201910304055Z), Key Research and Development Program of Jiangsu (Industry Prospects and Common Key Technologies) (BE2018093), Nantong Applied Research Project (JC2018115).

**Conflicts of Interest:** The authors declare no conflict of interest.

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
