# Peer review of "Study on the Coupling Relationship of Low Temperature Fluidity and Oxidation Stability of Biodiesel"

_applsci, doi:10.3390/app10051757_

Round 1

Reviewer 1 Report

In my view, all parts of this paper, including the Abstract, are written quite carelessly. I believe it must be re-submitted after a careful re-writing, preferably with the help of a native English speaker.

Reviewer 2 Report

Pag. 40 "The formation of macromolecular gelatinous substances causes problems such as fuel system gelation... ? How are gelatinous substances formed in biodiesel?.What is the composition of these substances?

Pag. 59 B1-B5.The B1 is not used?

Pag. 86 CFI is Bangjie diesel anti-coagulant. What is its composition?

Pag. 104 should be put Could Filter Plugging Point (CFPP)

In the CFI experiences, why don't the same amounts of diesel bangjie always be used for different diesel-biodiesel blends?

The addition of anti-coagulant would affect greenhouse gas emissions? To combustion engine performance by ignition?

Reviewer 3 Report

The manuscript presents interesting results, in terms of the use of biodiesel in mixtures with mixtures of fossil diesel, with the use of additives that improve the behaviour of the fuel, with respect to its flow properties at low temperatures. However the use of acronym of compounds and techniques is abused. This should be avoided, especially in the Abstract, where the acronym CFI, 0PD, BHA, PME and RME are handled, withoud defining, which prevents the reader from understanding the meaning of the described research.

Round 2

Reviewer 1 Report

I still feel that the article, including the abstract, should be re-written, perhaps with the help of a native English speaker. The paper can not be published in the present form.
